## PERSPECTIVE

# Getting excited about leaks: the atypical Na⁺ channel NALCN is a key determinant of native mouse anterior pituitary endocrine cell physiology

**Michael J. Shipston**[1,2]

[1]*Centre for Discovery Brain Science, Edinburgh Medical School: Biomedical Sciences, University of Edinburgh, Edinburgh, UK*
[2]*Zhejiang University-University of Edinburgh Joint Institute, Zhejiang University School of Medicine, Haining, PR China*

Email: mike.shipston@ed.ac.uk

The peer review history is available in the Supporting Information section of this article (https://doi.org/10.1113/JP288021#support-information-section).

Endocrine cells of the anterior pituitary (AP) regulate an eclectic array of physiological processes, from the control of growth, reproduction and metabolism to coordinating responses to stress. Half a century since the demonstration that native AP cells are electrically excitable, Belal et al. (2024) have revealed the atypical sodium 'leak' channel Nalcn as a key regulator of mouse AP cell resting membrane potential (RMP) and this spontaneous excitability (Fig. 1).

The majority of endocrine secretory cell types in the AP, including somatotrophs, corticotrophs and gonadotrophs, generate spontaneous calcium- dependent action potentials with typical durations of 20–50 ms (Fletcher et al., 2018). AP cells display a richness in the dynamics, regulation and pattern of electrical excitability that varies between different cell types: from spikes to complex pseudo-plateau bursting that are proposed to control hormone secretion. However endocrine AP cells have one feature in common – an unstable RMP (typical range −45 to −55 mV) that is considerably more depolarised (by >15 mV) than the equilibrium potential for potassium ($E_K$). Although considerable effort has focused on ion channels that control spiking and bursting, the molecular identity of ion channels that control this depolarised RMP has remained elusive – until now.

In AP endocrine cells replacement of external sodium ions (e.g. with NMDG⁺) results in significant hyperpolarisation and cessation of electrical excitability (Guerineau et al., 2021). In most pituitary cell types a significant depolarising linear inward current, carried by Na⁺ ions, is active at RMP that is typically inhibited by high external Ca²⁺ ($[Ca^{2+}]_e$) but insensitive to blockers of voltage-gated Na⁺ channels (e.g. TTX). Mathematical models of AP cell excitability require a depolarising inward 'leak' current to act against background K⁺ current to maintain the depolarised RMP – something, of course, predicted in the classic work of Hodgkin & Huxley published in this journal! These features are reminiscent of the atypical background Na⁺ current encoded by the evolutionary

**Figure 1. Nalcn controls RMP and spontaneous excitability of anterior pituitary cells**
Endocrine cells of the anterior pituitary are electrically excitable displaying a depolarised membrane potential significantly more positive than the equilibrium potential for potassium ($E_K$). A variety of 'core' conductances control the pattern of electrical activity of calcium-dependent spikes and bursts including voltage dependent calcium (Ca$_v$), potassium (K$_v$) and voltage- and -calcium activated potassium (BK) channels. Potassium and sodium leak currents oppose each other to determine the depolarised resting membrane potential. Genetic knockdown of the atypical Na channel Nalcn (Nacln$^{KD}$ cells) results in membrane hyperpolarisation and electrical silencing. Injection of a small depolarising current into Nalcn$^{KD}$ cells using dynamic clamp rescues electrical activity and the depolarised resting membrane potential.

conserved gene *NALCN* (Monteil et al., 2024).

With the lack of specific pharmacological inhibitors of Nalcn (Monteil et al., 2024). Belal et al. (2024) used a shRNA-knockdown strategy (Nalcn[KD]) using lentiviral transduction of unidentified dispersed native mouse AP cells as previously used in clonal GH3 cells (Impheng et al., 2021). In agreement with immunohistochemical analysis demonstrating widespread expression of Nalcn in the AP, electrophysiological analysis of Nalcn[KD] cells revealed a consistent (>90% of cells) hyperpolarisation (median 15 mV) and silencing of both spontaneous electrical activity and intracellular calcium transients compared to control cells. Importantly, normal RMP and spontaneous electrical activity was rescued in Nalcn[KD] cells using dynamic clamp to inject a surprisingly small depolarising conductance (0.02–0.16 nS), a value similar to the subtracted conductance required to silence control cells. A small depolarising current of just a few picoamps has a large effect on RMP as AP cells have a very high (>5 GΩ) input resistance. Furthermore, the NMDG-sensitive depolarising conductance in control cells (median 0.05 nS) was significantly reduced (>70%) in Nalcn[KD] cells suggesting this current is indeed mediated by Nalcn.

While providing fundamental new insight into the control of AP excitability this study also raises some intriguing questions: (i) is Nalcn the dominant depolarising leak current in all AP cell types, and how is its functional expression tightly regulated to control RMP? Significant variability in Nalcn current density is reported across the mixed AP population, and other channels such as TrpC have also been implicated (Fletcher et al., 2018; Guerineau et al., 2021). (ii) Does Nalcn assemble as a macromolecular complex with accessory subunits (Unc-79, Unc-80, Fam155A) (Monteil et al., 2024) in AP cells to control properties and regulation by hypothalamic hormones via their cognate GPCRs? (iii) Does Nalcn control hormone secretion across AP cell types? Although Nalcn[KD] in GH3 cells attenuated TRH-stimulated GH secretion (Impheng et al., 2021), the effect of removal of extracellular Na$^+$ on hormone secretion in other AP cells is not equivocal (Guerineau et al., 2021). Intriguingly, no consistent phenotype of modified AP hormone secretion has been reported in human or animal studies with mutations/deletion of Nalcn (Monteil et al., 2024). (iv) Does Nalcn control other calcium-dependent processes beyond hormone secretion such as gene expression, mRNA splicing or protein processing/degradation in AP cells? (v) Finally, Nalcn mRNA expression is dynamically regulated after a period of chronic stress in mouse corticotrophs (Duncan et al., 2024) suggesting dynamic control of Nalcn expression may be an important determinant of pituitary cell function in response to (patho)physiological challenges.

Understanding this small leak promises to reveal a fountain of knowledge of the physiological role of this intriguing channel in the anterior pituitary in health & disease.

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

## Additional information

### Competing interests

Michael Shipston: I work on the control of anterior pituitary corticotroph physiology and function.

### Author contributions

M.S.: Conception or design of the work; drafting the work or revising it critically for important intellectual content; final approval of the version to be published; agreement to be accountable for all aspects of the work.

### Funding

None.

### Keywords

anterior pituitary, cation channel, dynamic clamp, neuroendocrinology

### Supporting information

Additional supporting information can be found online in the Supporting Information section at the end of the HTML view of the article. Supporting information files available:

**Peer Review History**

