## [Peer Review History · The Journal of Physiology]

Getting excited about leaks: the atypical Na⁺ channel NALCN is a key determinant of native mouse anterior pituitary endocrine cell physiology

Michael J Shipston

DOI: 10.1113/JP288021

Corresponding author(s): Michael Shipston (mike.shipston@ed.ac.uk)

The following individual(s) involved in review of this submission have agreed to reveal their identity: Mino David Belle (Referee #1)

Review Timeline:

Submission Date:	14-Nov-2024
Editorial Decision:	13-Dec-2024
Revision Received:	13-Dec-2024
Accepted:	13-Dec-2024

Senior Editor: Peking Fong

Reviewing Editor: Yamuna Krishnan

Transaction Report:

Hi Peking

Thanks for turning this around so quickly.

I thought that - given the delay - it would be churlish to insist on sticking to 5 references so you'll see I've given the go-ahead for an additional one, if Mike would like to add it (see my 'ADDITIONAL NOTE' comment below).

As we had no response from Yamuna about this Perspective, I'll bypass her for the revised version.

All the best

Diana

Dear Dr Shipston,

Re: JP-P-2024-288021 "Getting excited about leaks: the atypical Na⁺ channel NALCN is a key determinant of native mouse anterior pituitary endocrine cell physiology" by Michael J Shipston

First of all, please accept our sincere apologies for the delay in providing you with an editorial decision on your Perspective.

Thank you for submitting your manuscript to The Journal of Physiology. It has been assessed by an Editor and by 1 expert referee and we are pleased to tell you that it is acceptable for publication following satisfactory minor revision.

The review comments are copied at the end of this email.

Your revised manuscript should be submitted online using the link in your Author Tasks <https://jp.msubmit.net/cgi-bin/main.plex>

LANGUAGE EDITING AND SUPPORT FOR PUBLICATION: If you would like help with English language editing, or other article preparation support, Wiley Editing Services offers expert help, including English Language Editing, as well as translation, manuscript formatting, and figure formatting at www.wileyauthors.com/eoo/preparation. You can also find resources for Preparing Your Article for general guidance about writing and preparing your manuscript at www.wileyauthors.com/eoo/prepresources.

REVISION CHECKLIST:

Check that your Methods section conforms to journal policy: <https://jp.msubmit.net/cgi-bin/main.plex>

Check that data presented conforms to the statistics policy: <https://jp.msubmit.net/cgi-bin/main.plex>

Upload a full Response to Referees file. To create your 'Response to Referees' copy all the reports, including any comments from the Senior and Reviewing Editors, into a Microsoft Word, or similar, file and respond to each point, using font or

background colour to distinguish comments and responses and upload as the required file type.

- 'Potential Cover Art' for consideration as the issue's cover image
- Appropriate Supporting Information (Video, audio or data set: see <https://jp.msubmit.net/cgi-bin/main.plex>)

We look forward to receiving your revised submission.

Yours sincerely,

Peying Fong
Senior Editor
The Journal of Physiology

EDITOR COMMENTS

You will see the Referee is most appreciative of your Perspectives piece. At this time, there are two minor comments that I leave for you to address at your discretion. I see that you conscientiously adhered to the guidelines for preparation of Perspectives pieces, limiting your references to five. Therefore, in order for you to address one of the Author's request, you would need to substitute one of the current references. Regarding the impact of making this substitution on the current manuscript's cohesiveness and integrity, I leave this for your judgement.

[ADDITIONAL NOTE FROM EDITORIAL OFFICE: HAPPY TO APPROVE AN ADDITIONAL REFERENCE HERE - SIX REFERENCES WILL BE FINE]

Many thanks for contributing this fine Perspectives article to The Journal of Physiology.

REFEREE COMMENTS

Referee #1:

I am delighted to read Prof. Michael Shipston's wonderful and very insightful Perspective article on our accepted manuscript, placing the potential importance of this work in a wider scientific context.

There are only two very minor comments for his consideration.

1) Page 3, line 47, please correct, Belal et al, not Bela et al.

2) We would be very grateful if the work of one of our coauthors on GH3 cells could be cited (Impheng et al., 2021, FASEB), if at all possible.

END OF COMMENTS

Response to referees

I am delighted the authors/reviewer found the perspective useful in highlighting this work.

I have addressed the minor concerns as below:

- 1) Page 3, line 47, please correct, Belal et al, not Bela et al.

My sincere apologies for the typo of first author - corrected

- 2) We would be very grateful if the work of one of our coauthors on GH3 cells could be cited (Impheng et al., 2021, FASEB), if at all possible.

I have included this citation in two place – in reference to KD strategy and as the primary reference for evidence that KD of Nalcn in GH3 cells affects secretion. As indicated by the editorial office they are happy to approve tis additional reference as I believe the other references and reviews are key to the perspective and for the wider J Physiology readership who will be very interested in this work

Dear Professor Shipston,

Re: JP-P-2024-288021R1 "Getting excited about leaks: the atypical Na⁺ channel NALCN is a key determinant of native mouse anterior pituitary endocrine cell physiology" by Michael J Shipston

We are pleased to tell you that your paper has been accepted for publication in The Journal of Physiology.

Yours sincerely,

Peying Fong
Senior Editor
The Journal of Physiology

If you would like to receive our 'Research Roundup', a monthly newsletter highlighting the cutting-edge research published in The Physiological Society's family of journals (The Journal of Physiology, Experimental Physiology, Physiological Reports, The Journal of Nutritional Physiology, and The Journal of Precision Medicine: Health and Disease), please click this link, fill in your name and email address and select 'Research Roundup':

<https://www.physoc.org/journals-and-media/membernews>

- You can help your research get the attention it deserves! Check out Wiley's free Promotion Guide for best-practice recommendations for promoting your work at: www.wileyauthors.com/eeo/guide. You can learn more about Wiley Editing Services which offers professional video, design, and writing services to create shareable video abstracts, infographics, conference posters, lay summaries, and research news stories for your research at: www.wileyauthors.com/eeo/promotion.

The Corresponding Author will receive an email from Wiley with details on how to register or log-in to Wiley Authors Services where you will be able to place an order

EDITOR COMMENTS

Many thanks for contributing this fine Perspectives article. It is now ready for final acceptance. I anticipate that it will draw broad interest, and will benefit The Journal of Physiology and the field at large. Congratulations!